# Alginate-Degrading Modes, Oligosaccharide-Yielding Properties, and Potential Applications of a Novel Bacterial Multifunctional Enzyme, Aly16-1

**DOI:** 10.3390/microorganisms12112374

**Published:** 2024-11-20

**Authors:** Lianghuan Zeng, Junge Li, Jingyan Gu, Wei Hu, Wenjun Han, Yuezhong Li

**Affiliations:** 1National Glycoengineering Research Center, Shandong Key Laboratory of Carbohydrate Chemistry and Glycobiology, NMPA Key Laboratory for Quality Research and Evaluation of Carbohydrate-Based Medicine, State Key Laboratory of Microbial Technology, Shandong University, Qingdao 266237, China; zenglianghuan2018@163.com (L.Z.); junge_li2020@hotmail.com (J.L.); hw_1@sdu.edu.cn (W.H.); lilab@sdu.edu.cn (Y.L.); 2United Post Graduate Education Base of Shandong University and Jinan Enlighten Biotechnology Co., Ltd., Jinan 250101, China; gujingyan76@aliyun.com

**Keywords:** alginate, degradation pattern, glycosaminoglycan, multifunctional, oligosaccharide preparation, polysaccharide lyase family

## Abstract

Relatively little is known about enzymes with broad substrate spectra, leading to limited applications and progress. Herein, we elucidate Aly16-1 of *Streptomyces* sp. strain CB16 as a novel multifunctional member of the eighth polysaccharide lyase (PL8) family, although it shared few sequence identities with the characterized enzymes. The recombinant enzyme rAly16-1 showed lyase activities against several acidic polysaccharides, including many glycosaminoglycan types, xanthan, and alginate. It was mannuronate (M)-preferred, endolytic, and optimal at 50 °C and pH 6.0. The smallest substrate was an ∆M-terminal (∆: unsaturated monosaccharide) trisaccharide, and the minimal product was ∆. In the final alginate digestions by rAly16-1, the fractions larger than disaccharides were ∆G-terminal (G: guluronate), while the disaccharides were mainly ∆M, showing an oligosaccharide-yielding property under the succession law. However, when degrading various oligosaccharides, rAly16-1 continued producing ∆M from the non-reducing end even when the substrates increased their sizes, quite different from the elucidated alginate lyases with variable alginate-degrading modes. Thus, co-determined by its M-preference, Aly16-1 is novel for its ∆M-yielding property in oligosaccharide preparations. Additionally, rAly16-1 can be applied in sequencing unsaturated trisaccharides, whether ∆M- or ∆G-terminal. This study provides novel insights into the characteristics and applications of a multifunctional enzyme within the PL8 family for resource explorations.

## 1. Introduction

Polysaccharide depolymerases (PDs) can catalyze the cleavage of glycosidic bonds within substrate chains, producing oligosaccharide products that are much more water-soluble and smaller in size, even monosaccharides. Therefore, PDs are important tools for humans to prepare functional oligosaccharides for further explorations and essential for bacterial strains to utilize polysaccharides as carbohydrate resources. According to their substrate spectra, PDs can be roughly divided into enzymes with a narrow spectrum or a broad one, even enzymes with multiple substrates. However, unlike well-known and widely applied enzymes such as *β*-mannanase [1], pullulanase [2], and cellulase [3], relatively little is known about multifunctional PDs, which has limited their broader enzymatic usage and further resource explorations.

Marine-derived polysaccharides and oligosaccharides have been identified to possess several important physiological functions, e.g., anticancer, antiviral, antioxidant, antiaging, glucose-lowering, hepatoprotective, and immune-regulatory effects [4,5]. Alginate, also known as sodium alginate, is one of the most important polysaccharide components of marine brown seaweeds [6,7] and is thought to play a role analogous to cellulose in terrestrial plants [8,9]. The alginate polysaccharide is linear and randomly composed of *β*-D-mannuronate (M) and its C5 epimer *α*-L-guluronate (G) through *β*-(1,4)-linked glycosidic bonds [10]. Alginate has been widely applied in pharmaceutical industries because of its good biocompatibility, water retention, antibacterial properties, and biodegradability [11]. In particular, G-enriched sugar chains can form strong gels after absorbing water [12]. Many reports have shown that the biological activity of an oligo-alginate chain is closely related to its degree of polymerization (DP), M/G ratio, and structural characteristics [13,14,15]. Recently, the enzymatic preparation of oligosaccharide fractions with designed structural characteristics has become the basis for sugar sequence analysis and for the direct preparation of oligo-alginate since the sale of GV-971 [16], an oligomannuronate-derived novel drug that aims to treat the early or middle stage of Alzheimer’s disease in China.

Alginate lyase can catalyze the degradation of alginate via a *β*-elimination reaction, newly forming a 4,5-unsaturated double bond at the non-reducing (*nr*) end of unsaturated oligosaccharide products [17,18]. These enzymes have been classified into several PL families in the Carbohydrate-Active Enzymes (CAZy; http://www.cazy.org) database, e.g., PL5, PL6, PL7, PL8, PL14, PL15, PL17, PL18, PL31, PL32, PL34, PL36, and PL39 [19,20,21,22]. Most characterized alginate lyases are endolytic enzymes that randomly cleave glycoside bonds within alginate chains, generating a series of size-defined unsaturated oligosaccharide fractions as the final products, and they usually possess relatively high and stable enzymatic activities. Therefore, endolytic enzymes are important tools for the preparation of various unsaturated oligosaccharide chains [7,23]. However, exolytic lyases can act on alginate chains to release monosaccharide units from the substrate chains for further biofuel production, i.e., they mainly release the unsaturated unit of ∆ and sometimes the saturated monosaccharide units of M or G for further bioconversion [24,25,26,27]. These enzymes usually exhibit relatively low enzyme activities and sensitive and unstable biochemical characteristics, although they are useful in preparing intermediate oligosaccharide product chains [25,27]. In recent reports, enzyme systems or multifunctional enzymes were found to be essential in digestion by animal gut bacteria because they can help enzymes with limited substrate spectra to further depolymerize diverse polysaccharides due to their broad spectra for better absorption by hosts [28]. Partial studies on alginate lyases have focused on the screening of enzymes with a narrow substrate spectrum or specific substrate selectivity, e.g., studies on the identification of enzyme-producing bacterial strains, modifications of enzymes to improve their activities and achieve better biochemical properties, and oligosaccharide-yielding properties and the corresponding mechanisms [29,30,31]. However, less is known about multifunctional PLs, except for a few reports on endo-type alginate lyases that contain various functional modules [32,33,34,35], which has limited their broader enzymatic usage and further applications. Thus, we are interested in multifunctional PLs with respect to their protein sequence information, substrate spectra, action models, and oligosaccharide-yielding properties for tool-like enzyme explorations.

Herein, we elucidate Aly16-1 of *Streptomyces* sp. strain CB16 as a novel multifunctional enzyme of the PL8 family, focusing on its sequence information, biochemical characteristics, and enzyme usage potentials.

## 2. Materials and Methods

### 2.1. Bacterial Strains, Carbohydrates, and Growth Conditions

Cells of *Escherichia coli* strains, i.e., DH5α or BL21 (DE3), were cultured at 37 °C for gene cloning or at 16 °C for protein production in Luria–Bertani (LB) broth, supplemented with ampicillin (at a final concentration of 100 μg/mL) or kanamycin (at 50 μg/mL) if necessary. The solid medium plate was prepared by supplementation with additional agar powder (1.5%, *w*/*v*).

Various polysaccharides, i.e., agarose, alginate (viscosity: ≥2000 cP; 2% (25 °C)), chondroitin, chondroitin sulfates (CS-A, C, D, and E types), hyaluronan (HA), heparin, heparin sulfate (HS/DS), pectin, and xanthan, were purchased from Sigma–Aldrich Co., Ltd., USA. Poly-G blocks, poly-M blocks, and standard size-defined G-enriched or M-enriched saturated alginate chains (ranging from disaccharide to heptasaccharide fractions, with promised purities of >95%) were purchased from Qingdao Biozhi Biotech Co., Ltd., Qingdao, China. Various size-defined unsaturated oligosaccharide fractions were prepared using recombinant alginate lyases, i.e., the M-preferred exolytic alginate lyase rAly6 of *Flammeovirga* sp. strain MY04 and the M-specific endolytic alginate lyase Pae-rAlgL of *Pseudomonas aeruginosa*, as described in a previous study [27], or using the recombinant alginate lyase rAly16-1.

### 2.2. Gene Cloning and Sequencing

The DNA sequence of ORF03870 (GenBank: WP_073960516), encoded by the genome of *Streptomyces* sp. strain CB16 (CGMCC no. 12351), was translated into the amino acid sequence of the predicted protein Aly16-1 using the BioEdit software version 7.02 [36]. The signal peptide was identified using the SignalP online server 5.0 (http://www.cbs.dtu.dk/services/, accessed on 22 September 2024). The molecular weight and the isoelectric point (pI) of the predicted protein were estimated using the peptide mass tool on the ExPASy server of the Swiss Institute of Bioinformatics (http://swissmodel.expasy.org/, accessed on 22 September 2024). Online similarity searches of the protein sequence were performed using the BLAST algorithm on the National Center for Biotechnology Information server (http://www.ncbi.nlm.nih.gov, accessed on 22 September 2024) and the SMRT online server (https://smart.emble.de, accessed on 22 September 2024). Multiple-sequence alignments and phylogenetic analyses were performed using the MEGA software version 7.2.5 [37].

The genomic DNA of *Streptomyces* sp. strain CB16 was prepared and purified using the FastPure bacterial DNA isolation mini kit (Vazyme Biotech Co., Ltd., Nanjing, China). To obtain the recombinant protein rAly16-1, the full-length gene of Aly16-1 was amplified from the genomic DNA of the CB16 strain using high-fidelity Vazyme™ LAmp DNA PolyMerase (Vazyme Biotech Co., Ltd., Nanjing, China) and the primers aly16-1-F (5′-gcatatgTCCGGCACGGCACGGACCG-3′) and aly16-1-R (5′-gtctagaGCTCCTCTTCAGGGTGA GGCC-3′), which were designed for the restriction enzyme sites (underlined) of *Nde* I and *Xba* I in the plasmid pET-30a (Invitrogen, Shanghai, China), to fuse a 6 × His tag at the *C*-terminus of the recombinant protein. The PCR product was subsequently gel-recovered and finally ligated into the enzyme-digested vector pET-30a by the T4 DNA ligase (TaKaRa, Dalian, China), producing the resultant plasmid pET-30a-Aly16-1 to yield the recombinant protein rAly16-1.

### 2.3. Heterologous Expression and Protein Purification

*Escherichia. coli* BL21 (DE3) cells harboring the plasmid, e.g., pET-30a-Aly16-1, were initially cultured in 100 mL of LB broth containing 100 µg/mL of ampicillin and shaken for ~2 h at 37 °C at 200 rpm. When the cell density reached an *A*_600_ value of 0.8, the cells were then induced to start targeting protein expression by supplementation with isopropyl 1-thio-β-D-galactoside (IPTG), the inducer, at a final concentration of 0.05 mM with shaking at 16 °C for 24 h at 220 rpm. The cells were harvested by centrifugation at 10,000× *g* for 10 min, washed twice using ice-cold buffer A (50 mM Tris and 150 mM NaCl; pH 8.0), resuspended in buffer A, and disrupted by sonication (80 repetitions; 5 s). After centrifugation at 15,000× *g* for 30 min, the supernatant containing each soluble recombinant protein was loaded into a buffer A-equilibrated Ni-nitrilotriacetic acid agarose (Ni-NTA) column (TaKaRa, Dalian, China). Subsequently, the protein-bound column was eluted using buffer A, which each contained imidazole in gradient concentrations, i.e., 0, 10, 20, 50, and 250 mM. The fractionated protein samples were finally analyzed using SDS–PAGE. To obtain an active polysaccharide depolymerase, the purified protein fractions were dialyzed against buffer B (50 mM Tris; 50 mM NaCl; 5% glycerol (*v*/*v*); pH 8.0).

SDS–PAGE was performed using 13.2% (*w*/*v*) polyacrylamide gels according to the method of Sambrook and Russell [38]. The proteins were detected by staining gels with Coomassie Brilliant Blue R-250. The protein concentrations were individually determined by the Folin–Lowry method using Folin–Ciocalteu’s phenol reagent (Sigma–Aldrich, Burlington, MA, USA), with bovine serum albumin as the standard [38].

### 2.4. Enzyme Activity Assay Toward Various Substrates

To determine the substrate spectrum of rAly16-1, various polysaccharides were used as testing substrates and individually dissolved in deionized water to prepare stock solutions (3 mg/mL). Each stock solution (100 µL) was mixed with 100 µL of appropriately diluted enzyme and 100 µL of NaAc-HAC buffer (50 mM; pH 6.0) and then incubated at 50 °C for 72 h. The enzyme-treated samples were heated in boiling water for 10 min and subsequently ice-cooled. After centrifugation at 15,000× *g* for 10 min, each supernatant was collected and analyzed by measuring the absorbance at 540 nm using the 3,5-dinitrosalicylic acid (DNS-reducing sugar) method [39], where 1 unit was defined as the amount of enzyme required to release 1 µmol of the reducing sugars per minute under the optimal reaction conditions.

Subsequently, each product was observed at 235 nm via gel filtration HPLC and then purified into the same size-defined oligosaccharide fractions. HPLC chromatography by a Superdex^TM^ 30 Increase 10/300 GL (GE, Cincinnati, OH, USA) size exclusion column was performed using 0.20 mol/L of NH_4_HCO_3_ at the speed of 0.40 mL/min [40,41]. Finally, each final oligosaccharide product fraction was collected, frozen, and repeatedly dried to remove water and the salt NH_4_HCO_3_, and the chemical structure of each product fraction yielded by rAly16-1 was further determined using ^1^H-NMR (nuclear magnetic resonance) spectroscopy with a JNM-ECP600 (JEOL, Kyoto, Japan) instrument set at 600 MHz.

### 2.5. Biochemical Characterization

The optimal temperature was determined using the alginate polysaccharide as a substrate in 50 mM NaAc-HAc buffer (pH 6.0) at temperatures ranging from 0 to 70 °C for 1 h. The experiments were performed in a total volume of 300 µL, with 30 µL of approximately diluted enzymes. To determine the thermostability of the recombinant proteins, the residual enzyme activities were measured after each enzyme was incubated at various temperatures, ranging from 0 to 70 °C, for 0–24 h. The optimal pH value was determined using the following buffers with different pH values: 50 mM NaAc-HAc buffer (pH 5.0–6.0), 50 mM NaH_2_PO_4_-NaHPO_4_ buffer (pH 6.0–8.0), or 50 mM Tris-HCl buffer (pH 7.0–10.0, adjusted at 4 °C). The effects of the different pH values on the enzyme stability were determined by measuring the residual activity after incubating rAly16-1 at 4 °C at various pH values (5.0–10.0) for 2 h. The effects of metal ions and chelating agents on alginate-degrading activities were examined by determining the enzyme activity of each reaction in the presence of 1 mM or 10 mM concentrations of the various chemicals, respectively. All reactions were performed in triplicate. After each treatment, the enzyme activity was estimated using the DNS-reducing sugar method [39].

### 2.6. Analyses of Carbohydrate Substrate Action Modes

The digestion by rAly16-1 was first performed using the alginate polysaccharide as a substrate (3 mg/mL) at 50 °C for over 72 h. To further determine the oligosaccharides in each final product fraction, at appropriate time intervals, the incubation reaction mixtures were heated in boiling water for 10 min, subsequently cooled to 4 °C, and centrifuged at 15,000× *g* for 10 min. Finally, the resultant supernatants were individually transferred for size-exclusion HPLC analyses, as previously described.

To further discover the oligosaccharide-degrading patterns of Aly16-1, various standard size-defined M-enriched or G-enriched saturated sugar chains were individually reacted with the purified enzyme. To analyze the catalytic mechanism of Aly16-1, unsaturated oligosaccharide fractions with differing sizes (DPs), i.e., intermediate oligosaccharide products of ∆G-terminal UDP2, UDP3, UDP4, UDP5, and UDP6 fractions yielded by rAly6 [27], were each used as testing substrates. The final UDP3 product fractions, containing both ∆M-terminus and ∆G-terminus (*w*/*w*, ~1:1) fractions, which were yielded by the endolytic alginate lyase Pae-rAlgL [27], were also used as testing substrates.

To determine the action pattern of rAly16-1 further, various size-defined saturated oligosaccharide fractions were fluorescently labeled at their reducing (r) ends using excess 2-AB (Sigma–Aldrich, USA) [40]. Then, the labeled products (~1 µg each) were individually purified by gel filtration HPLC and then further degraded using rAly16-1. Each reaction product was finally analyzed using a fluorescence detector, with excitation and emission wavelengths of 330 and 420 nm, respectively.

## 3. Results

### 3.1. Sequence Information of Aly16-1

The genome of the polysaccharide-degrading bacterium *Streptomyces* sp. strain CB16 contained a putative polysaccharide lyase gene, named ORF03870 (GenBank: PQ333014), which was 2418 bp in full length and encoded the protein Aly16-1. The molecular mass of Aly16-1 was ~85.3 kDa, with an isoelectric point (pI) of 6.1, and the *N*-terminal amino acid (Met^1^ to Ala^38^) was a type I signal peptide.

SMART online analysis of the domain composition and organization showed that the protein Aly16-1 contained a putative *N*-terminal module (Trp^57^ to Gln^390^) of the lyase_8_N module, followed by a putative lyase_8 module (Asn^429^to Ser^684^), and a putative lyase_8_C module (Glu^698^-Val^762^) (Figure 1A) [42,43]. While according to the newest CAZy database (downloaded on 22 September 2024), BLASTp (Global Alignment) similarity searches showed that the protein Aly16-1 shared a sequence identity below 30% with a total of 23 characterized PL8 enzymes, including hyaluronan lyases, chondroitin sulfate lyases, and alginate lyases, e.g., Psman8A of the fungal strain *Paradendryphiella salina* (GenBank: VYG66350.1), it is the first enzyme (M-specific alginate lyase) reported within the PL8 family [19,42,44]. Therefore, the Aly16-1 protein was predicted to be an enzyme with the exact function to be discovered.

A single band of the recombinant protein rAly16-1, fused with a *C*-terminal 6 × His tag, was purified from *E. coli* strains at an imidazole concentration of 50 mmol/mL, which was consistent with the predicted molecular weight in the SDS–PAGE gel (Figure 1B).

### 3.2. Broad Substrate Spectrum

Initially, a total of nineteen different polysaccharides were used as testing substrates in the DNS-reducing sugar tests to investigate the substrate preference of rAly16-1. Subsequently, approximately 1.0~2.0 μg of the production mixture of each hypothetical polysaccharide substrate was further examined by a gel filtration HPLC analysis and ~1.0 mg for each ^1^H-NMR test. As a result, seven types of acidic polysaccharide substrates yielded reducing sugars in the reacting system. Furthermore, when compared with the control groups, the specific signals of H-absorbance at 5.6~5.7 ppm were significantly enhanced in individual digestion, suggesting that the Aly16-1 enzyme can digest multiple acidic polysaccharides, i.e., many glycosaminoglycans types (i.e., obviously acting on the HA, CS-A, CS-C, and CS-E polysaccharides, whereas it acted weakly on HS/DS), xanthan, and particularly alginate (a stronger 5.6~5.7 ppm H-absorbance) (Appendix A), suggesting it to be a lyase via the similar *β*-elimination mechanism.

### 3.3. Biochemical Characteristics

The enzymatic characteristics of rAly16-1 were further determined using alginate as a polysaccharide substrate. The recombinant protein rAly16-1 showed the highest activity at 50 °C (Appendix A) and retained more than 80% of its original activity after incubation at temperatures from 0 to 30 °C for 24 h (Appendix A). The optimal pH value of rAly16-1, determined at 50 °C in 50 mM NaAc-HAc buffer, was 6.0 (Appendix A). Furthermore, the enzyme activities of rAly16-1 were strongly inhibited by Ag^+^, Cu^2+^, Hg^2+^, Zn^2+^, Fe^3+^, and ethylenediamine tetra-acetic acid (EDTA) (Appendix A).

### 3.4. Alginate-Degrading Patterns

Under the optimal condition of 50 °C in 50 mM NaAc-HAc (pH 6.0) for 30 min, rAly16-1 showed much higher activity against poly-M blocks (~15,284 U/mg) than against poly-G blocks (~438 U/mg), while it showed an activity of only ~2134 U/mg against alginate, suggesting that Aly16-1 is obviously M-preferred. As the enzymatic reaction progressed, rAly16-1 produced much more oligosaccharide products of alginate and poly-M blocks, whereas it produced a lower proportion of poly-G blocks (Figure 2A), which confirmed the M-preference of the Aly16-1 enzyme.

To investigate the degradation pattern of rAly16-1, the alginate polysaccharide was enzymatically reacted for over 72 h, and then each resultant product was removed at appropriate time intervals and analyzed at 235 nm by size-exclusion HPLC chromatography. Initially, the main product fractions were unsaturated oligosaccharides with high DPs, which were then gradually converted into smaller oligomers and, finally, into a series of size-defined oligosaccharide fractions (Figure 2B), indicating that Aly16-1 is a typical endolytic alginate lyase. However, the peak at 20.6′ min demonstrated that oligo-alginate chains with high DPs existed from the beginning to the end of the reaction (Figure 2B), suggesting that Aly16-1 cannot completely degrade the alginate polysaccharide, possibly due to the M-preference of Aly16-1 and the G-composition of the alginate substrate (Figure 2A).

Moreover, as shown in Figure 2A,B, rAly16-1 showed a novel oligosaccharide-yielding property, as the final main unsaturated oligosaccharide product of rAly16-1 retained its disaccharide size of ∆M whenever the alginate polysaccharide or the poly-M blocks were used as the substrate. Thus, we were interested in the disaccharide-yielding property of rAly16-1, i.e., what the unsaturated disaccharide products really were and the associated mechanisms via which they were produced.

### 3.5. Degradation Pattens of Saturated Oligosaccharides and 2AB-Labeled Derivatives

When reacted with saturated alginate oligosaccharides and investigated at 235 nm by size-exclusion HPLC, rAly16-1 did not digest any M-enriched disaccharide M2. Meanwhile, it could degrade a larger substrate chain, e.g., M3, a saturated M-enriched trisaccharide, to produce a UM2 (i.e., ∆M, an unsaturated disaccharide) product fraction (from the *r*-end) and a hypothetical saturated monosaccharide product unit M (from the *nr*-end) in a low proportion. Furthermore, rAly16-1 could incompletely digest the M4 substrate chain (an M-enriched and saturated tetrasaccharide) into equimolar UM2 (∆M from the *r*-end) and M2 (a saturated disaccharide from the *nr*-end) chains (Figure 3A). Moreover, rAly16-1 yielded UM2 or UM3 (an M-enriched and unsaturated trisaccharide) units from the *r*-end of M5 (an M-enriched and saturated polar pentasaccharide) substrate chains, with high degradation molar ratios of 75.8% and 24.1%, respectively (Figure 3A,C), thereby leading to the final main product being a UM2 fraction. Similarly, there were three action modes of the M6 substrate chains (saturated hexasaccharides) when reacted with rAly16-1, yielding M2, M3, and M4 products (from the *nr*-end), whereas it continued producing ∆M (UM2 from the *r*-end) as the main unsaturated oligosaccharide product up to a ~81.1% degradation molar ratio (Figure 3C).

Thus, the M-preferred endolytic enzyme Aly16-1 has a novel oligosaccharide-yielding property, as the final main unsaturated oligosaccharide product of rAly16-1 retained its disaccharide size of ∆M and the yielding from the r-end, even when M-enriched saturated substrate chains enlarged their sizes (Figure 3A), which is theoretically similar to the degradation of natural algal-derived alginate polysaccharide.

Furthermore, rAly16-1 did not degrade any G-enriched saturated oligosaccharide chains (Figure 3B), indicating that Aly16-1 is an M-specific endolytic polysaccharide lyase, perhaps due to the insufficient purity of the poly-G blocks formerly described in the DNS-reducing sugar tests in this study.

Notably, the recombinant enzyme rAly16-1 could hardly digest the saturated disaccharide M2 or the saturated trisaccharide M3 fractions, even after they were labeled at their reducing ends by 2AB, i.e., 2AB-M2 or 2AB-M3. Meanwhile, it could partially degrade larger size-defined saturated chains, e.g., the 2AB-M4 chain, producing 2AB-M2 (from the *r*-end) and yielding ∆M chains (from the *nr*-end) as the final main product (Figure 4A). Oligosaccharide substrate fractions of 2AB-M5 and 2AB-M6 were also partially degraded by rAly16-1, while their main products from the *r*-end were the same as 2AB-M3 in size (Figure 4B,D). In addition, rAly16-1 could hardly degrade any large-sized 2AB-labeled, saturated, and G-enriched oligosaccharide testing substrates, e.g., 2AB-G5 and 2AB-G6 substrate chains (Figure 4C), which further confirmed that Aly16-1 is M- rather than G-preferred. 

### 3.6. Oligosaccharide-Yielding Properties of rAly16-1

To identify the structures of the final oligosaccharide products of alginate by rAly16-1, unsaturated disaccharide (UDP2) to unsaturated octasaccharide (UDP8), seven size-defined oligosaccharide fractions were individually purified from the alginate digestion (Figure 5A). According to existing reports [7,25], the structures of sugar residues next to ∆ units can be directly identified based on the H-4 signals of the ∆ units in the ^1^H-NMR data of unsaturated oligosaccharides, e.g., H-4∆M or H-4∆M. In the case of UDP2-UDP8 product fractions, specific signals at ~5.63 ppm of the H-4 of the ∆G units were observed only in a series of size-defined unsaturated product fractions that were larger than a UDP2’s size. Meanwhile, interestingly, much stronger H-4 signals of the ∆M units (~5.68 ppm) were particularly observed in the DP2 product fractions, a disaccharide size, indicating that the final main disaccharide products were predominantly ∆M units (Figure 5B).

Further HPLC analyses also showed that rAly16-1 did not degrade small size-defined unsaturated oligosaccharides that were yielded by the M-preferred and exolytic alginate lyase rAly6 [27], i.e., the ∆G-terminal trisaccharide intermediate product fraction of UDP3 (∆GX, X = M or G), whereas it could digest larger size-defined substrate chains, e.g., a ∆G-terminal tetrasaccharide fraction of UDP4 (∆GXX) into two UDP2 molecules or equimolar UDP3 and ∆ product fractions (Figure 6A), although the former digestion indicated only a 21% degradation ratio according to the peak area integrals. However, rAly16-1 could degrade the final UDP3 product chains produced by the M-specific endolytic enzyme Pae-rAlgL (∆MX/∆GX; ~1:1) into equimolar UDP2 and ∆ fractions, with a larger degradation ratio of 45% (Figure 6B). Although the substrate chains were similar in the same trisaccharide size, the degradation ratios were higher and quite different, mainly due to a higher preference for ∆M-terminal than ∆G-terminal chains [27]. Thus, the enzyme Aly16-1 can be applied in sequencing the *nr*-end of an alginate-derived unsaturated trisaccharide fraction, i.e., to determine whether it is ∆M- or ∆G-terminal.

Furthermore, the smallest substrate of Aly16-1 was determined to be an M-enriched unsaturated trisaccharide, whereas the minimal product was an unstable monosaccharide unit of ∆ (Figure 3A,B and Figure 6A,B).

In further digestions, it was indicated that larger unsaturated oligo-alginate fraction substrates, e.g., UDP5, UD6, and larger chains produced by rAly6, an M-preferred and exolytic alginate lyase [27], could be only partially degraded by the M-preferred and endolytic lyase rAly16-1 (Figure 6A), even when the enzyme usage and the reaction time were excessive. Therefore, the case was that the tested substrate chains, intermediate oligosaccharide products yielded by Aly6, contained both M and G units inside. Meanwhile, interestingly, in each final digestion by rAly16-1, the largest unsaturated product chains remained smaller than the corresponding substrates by the disaccharide size (Figure 6A), while the predominant products were further determined to be ∆M units in the HPLC analyses and ^1^H-NMR tests, the same as shown in Figure 2A and Figure 5B, indicating that the enzyme Aly16-1 has a novel oligosaccharide-yielding property, i.e., disaccharide-yielding, as a PL8 alginate lyase [19,27,45,46].

Thus, combining the M-preference, we assumed that the enzyme Aly16-1 is preferred to recognize and degrade possible sites (indicated by ambiquous arrows) of M-enriched motifs within an alginate chain, e.g., M-MMXn, G-MMXn, or ∆-MMXXn (−, glycosidic bond; Figure 6C).

## 4. Discussion

The gene ORF03870 within the genome of the polysaccharide-degrading bacterium *Streptomyces* sp. strain CB16 encodes a PD named Aly16-1. This protein can be classified into the PL8 family for containing potential functional modules (Figure 1A). Generally, reported lyases containing a GAG module can act on various polysaccharide substrates, e.g., HA and CS-A, CS-C, CS-D, and CS-E types, while few are active against other polysaccharide types besides the GAG types [45,46,47,48,49,50]. In this study, although sharing sequence identities below 30% with each characterized enzyme in the database of the PL8 family, e.g., GAG lyases or alginate lyases, the enzyme rAly16-1 showed degradation activities against various polysaccharides in DNS-reducing sugar tests, and *β*-elimination digestions were further confirmed by ^1^H-NMR tests (Appendix A). Therefore, the protein Aly16-1 is defined as a novel PL8 member, in addition to being multifunctional toward many GAG types and also active against non-GAG polysaccharides, such as xanthan and alginate. In further tests using the purified recombinant protein rAly16-1 (Figure 1B), this muti-activity also led to a wide range of polysaccharide digestions by the lyase Aly16-1, yielding corresponding unsaturated oligosaccharides for more explorations.

Using alginate as a polysaccharide substrate, rAly16-1 was determined to be M-preferred, initially by DNS-reducing sugar tests of different substrates and, finally, by HPLC analyses of various oligosaccharide digestions (Figure 2A,B and Figure 3A,B). Moreover, rAly16-1 was determined to be endolytic by HPLC analyses (Figure 2A) and was biochemically characterized by experiments, e.g., it was optimal at 50 °C and pH 6.0, respectively (Appendix A). In addition, by further reactions and analyses, the smallest oligo-alginate substrate of rAly16-1 was determined to be the ∆M-terminal trisaccharide, while the minimal product was the theoretical ∆ unit (Figure 3A,B and Figure 6A,B).

Interestingly, in the ^1^H-NMR tests, a series of size-defined final oligosaccharide products yielded by rAly16-1 showed a typical succession law (Figure 3A,B), i.e., turning from ∆G-ended and G-enriched large oligosaccharides product fractions (larger than or equal to trisaccharides) into unsaturated disaccharides of mainly ∆M, indicating the M-preferred and endolytic variable action mode of Aly16-1 as an alginate lyase, which was quite similar to the M-specific endolytic alginate lyase Pae-rAlgL of *P*. *aeruginosa* [27], whereas it was quite different from elucidated G-preferred or bifunctional but G-preferred alginate lyases [45,46]. Therefore, Aly16-1 is novel for being capable of preparing ∆M disaccharides. Furthermore, we accordingly summarized two possible mechanisms that are closely associated with its ∆M-producing property: the M-preference and the special disaccharide-yielding mode. Additionally, an image of the possible cleavage motif of Aly16-1 was also provided (Figure 6C).

As known, PDs with broad substrate spectra, including GAG lyases, are not as widely applied as PDs with narrow ones, particularly in learning the relationships between the structures and functions of oligosaccharides, possibly due to their random cleavages against polysaccharide substrates and corresponding yields of oligosaccharide products. Therefore, PDs sold on the market are generally enzymes with narrow spectra. In this study, we explored a novel application of the multifunctional enzyme rAly16-1, i.e., determining whether it was an ∆M-end or a ∆G-end of a trisaccharide chain, by corresponding digestions and further HPLC analyses (Figure 6B). Thus, the multifunctional lyase Aly16-1 can be applied in oligosaccharide sequencing in an initial step, which was also co-determined by its M-preference and substrate-degrading mode.

Hence, novel insights into multifunctional enzymes will benefit wider enzyme applications and further resource explorations, particularly in the direct oligosaccharide preparation of ΔM and improvement of associated oligosaccharide-sequencing technologies.

## 5. Patent

The multifunctional polysaccharide depolymerase Aly16-1, with its M-preference against alginate, corresponding oligosaccharide-yielding properties, and potential oligosaccharide preparation and sequencing applications, was authorized by the China invention patent under No. ZL 2021 1 0379188.

## Figures and Tables

**Figure 1 microorganisms-12-02374-f001:**
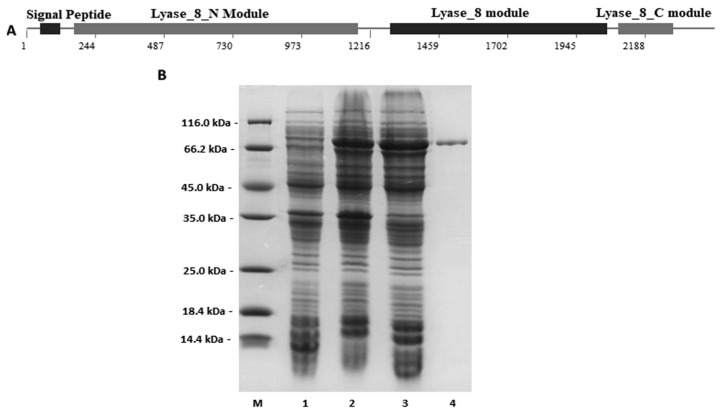
Sequence properties of the protein Aly16-1 encoded by the *Streptomyces* sp. strain CB16 and purification of the recombinant protein. (**A**) Modular architecture of the protein Aly16-1; the numbers indicate the corresponding nucleotides of the encoding gene. (**B**) Protein purification and SDS-PAGE analysis; lane M, unstained protein molecular weight marker product# 26,610 (Thermo Scientific, Waltham, MA, USA); lane 1, the induced cell lysate of the negative control group; lane 2, the induced cell lysate of rAly16-1; lane 3, the supernatant fluid containing rAly16-1; lane 4, the purified protein of rAly16-1.

**Figure 2 microorganisms-12-02374-f002:**
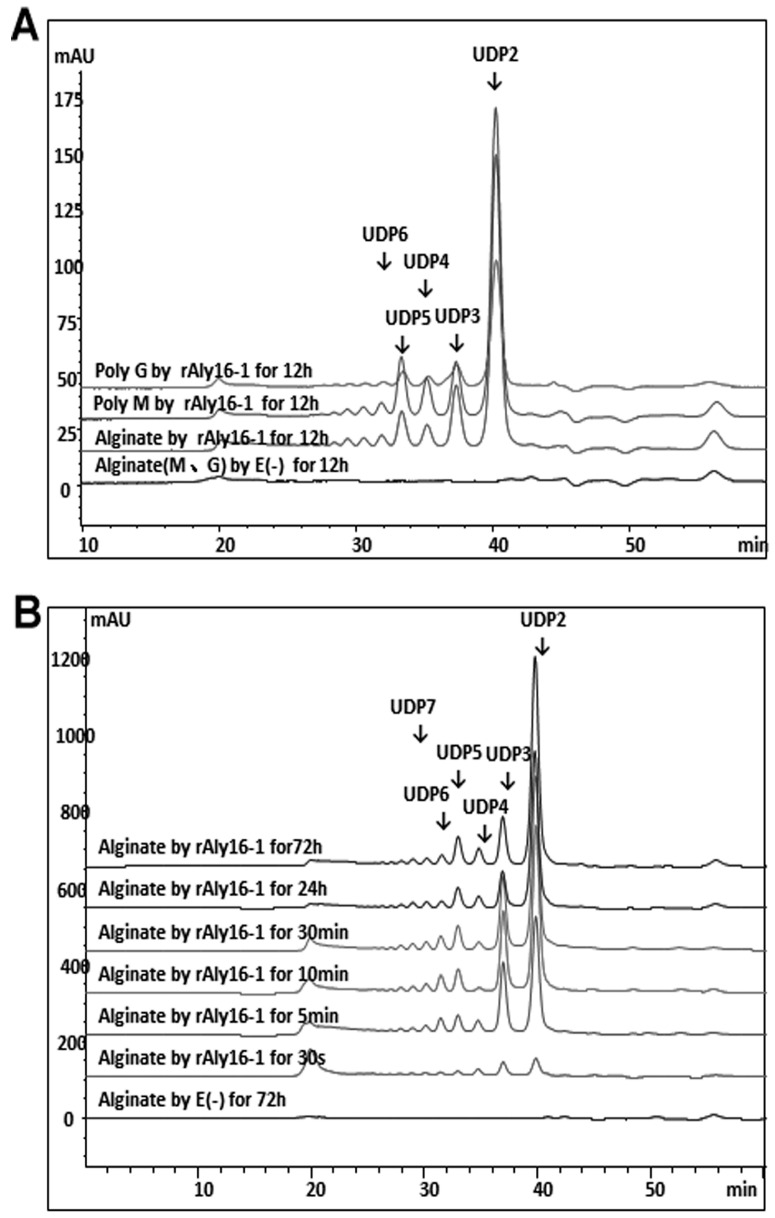
HPLC analyses of alginate chains degraded by rAly16-1. (**A**) Alginate, poly-M, and poly-G block substrates digested by rAly16-1. (**B**) Time-course of alginate digested by rAly16-1. E (–), negative control group treated with the inactivated enzyme.

**Figure 3 microorganisms-12-02374-f003:**
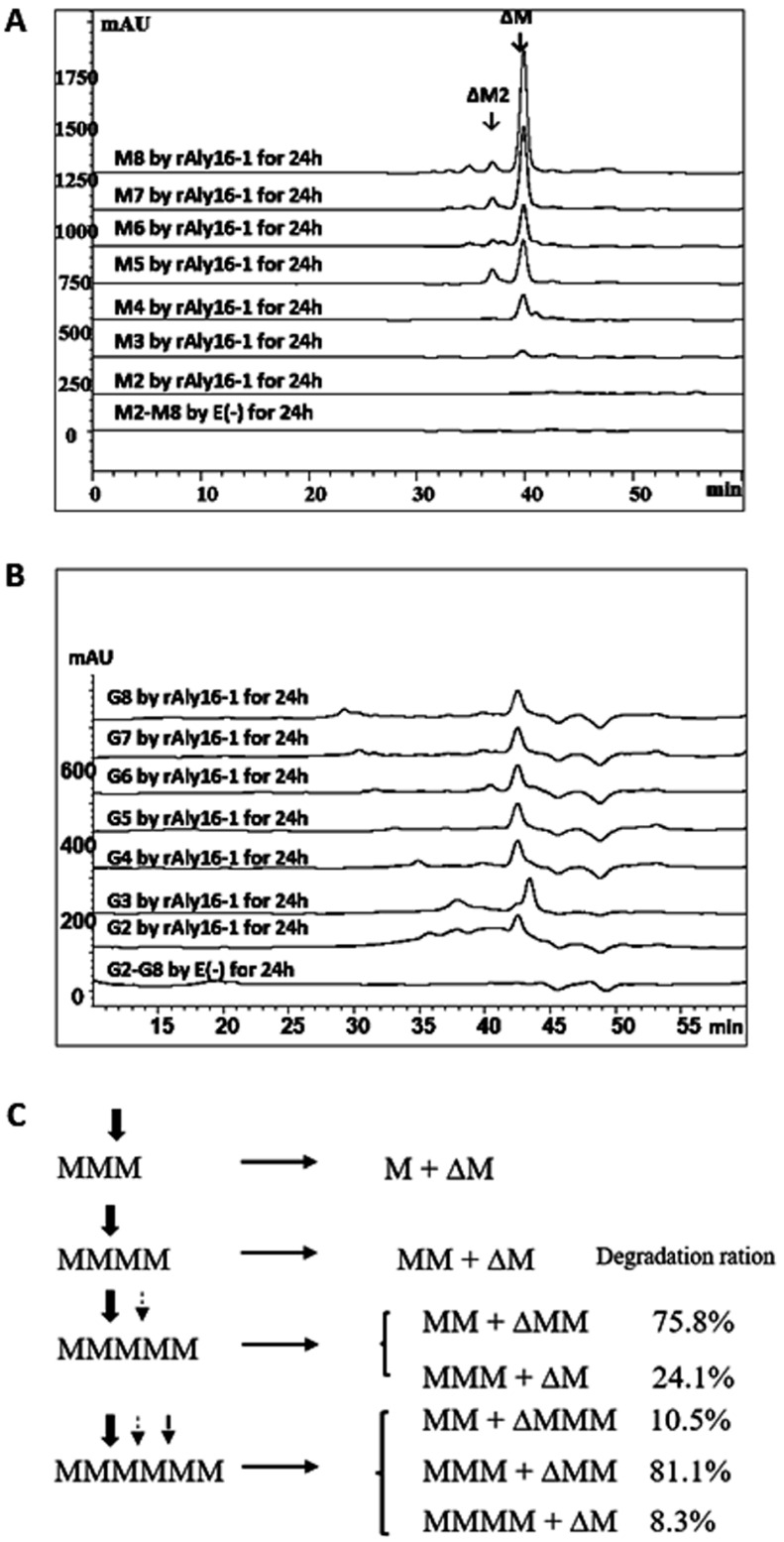
HPLC analyses of saturated oligosaccharide chains digested by rAly16-1 and possible enzymatic mechanisms. (**A**) Testing substrates of M2~M8 chains; (**B**) testing substrates of G2~G8 chains; (**C**) degradation pattern diagrams of M3~M6 chains by rAly16-1. Different arrows indicate different possible cleavge sites of Aly16-1 within the substrate chain.

**Figure 4 microorganisms-12-02374-f004:**
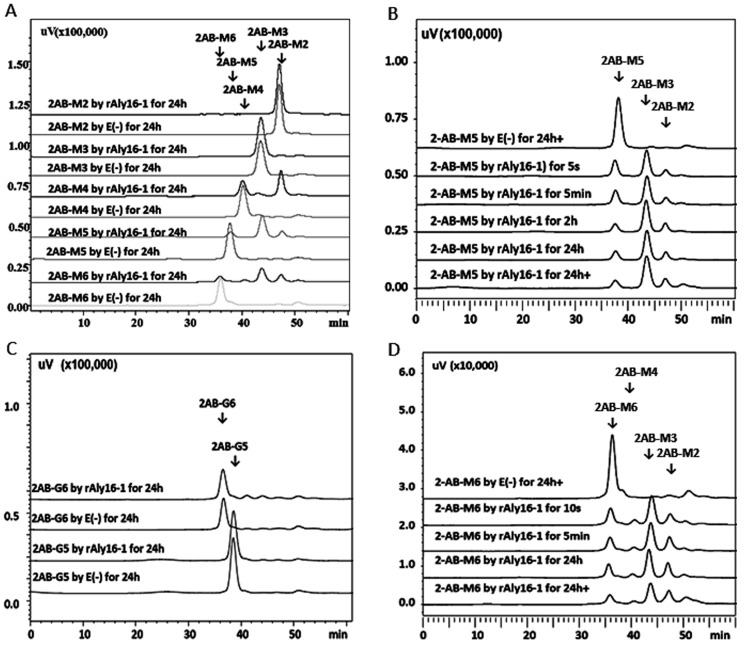
Fluorescent HPLC analyses of saturated oligosaccharide substrate chains reacted with rAly16-1. (**A**) Final products of 2AB-labeled M2~M6 substrate chains by rAly16-1; (**B**) time-course digestion of the 2AB-M5 chain by rAly16-1; (**C**) time-course digestion of 2AB-G5 and 2AB-G6 chains by rAly16-1; (**D**) time-course digestion of 2AB-M6 by rAly16-1. E (−), negative control group treated with the inactivated rAly16-1 enzyme.

**Figure 5 microorganisms-12-02374-f005:**
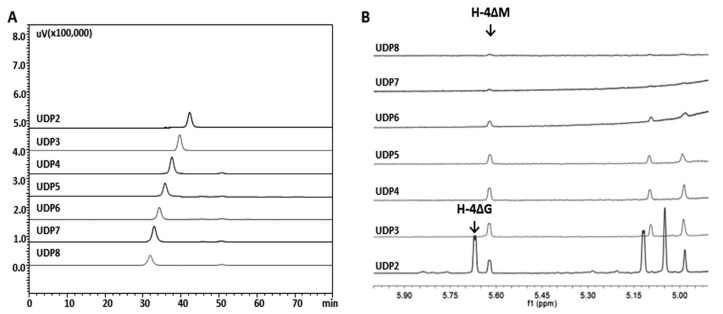
The ^1^H-NMR data of each size-defined final oligo-alginate product fraction yielded by rAly16-1. (**A**) Individually purified UDP2-UDP8 oligosaccharide product fractions were gel filtration HPLC-analyzed. (**B**) The ^1^H-NMR data of each size-defined oligosaccharide fraction.

**Figure 6 microorganisms-12-02374-f006:**
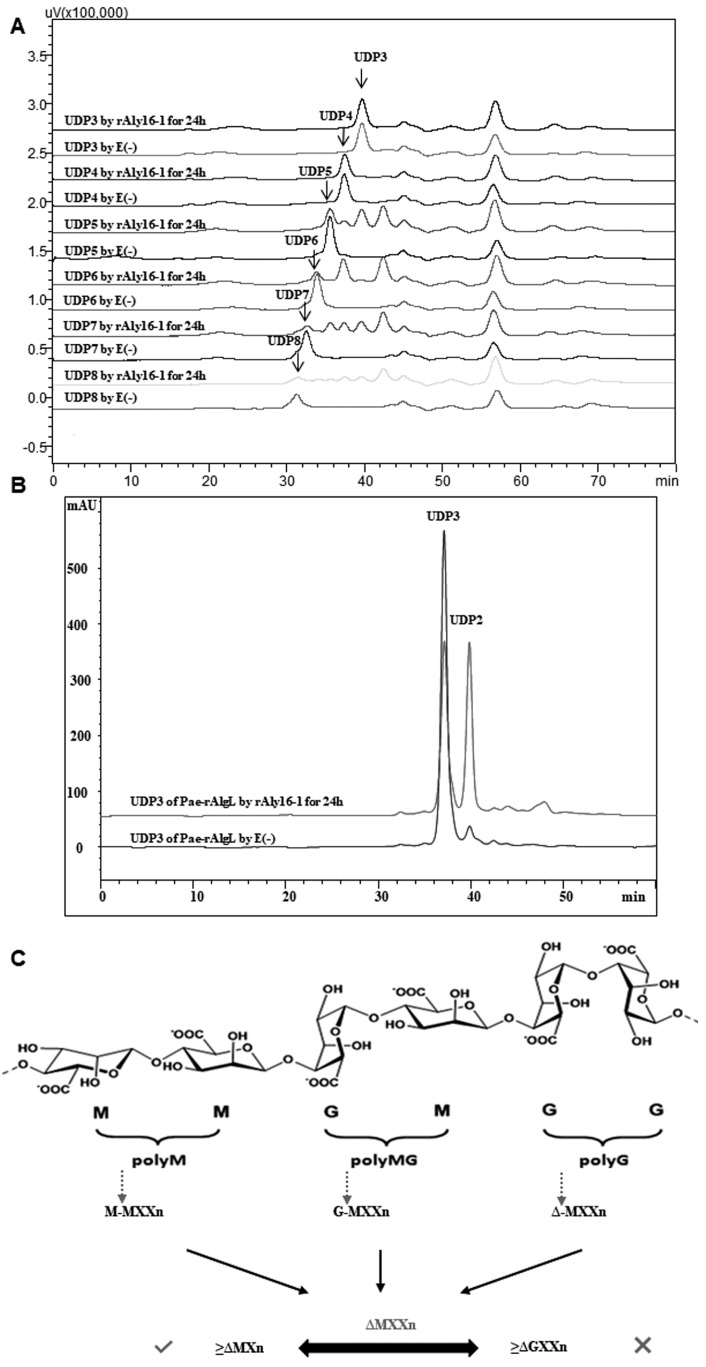
HPLC analyses of final products of unsaturated oligosaccharides reacted with rAly16-1. (**A**) Various ∆GXn substrate chains produced by rAly6 were digested by rAly16-1. (**B**) ∆GX/∆MX chains produced by Pae-rAlgL were digested by rAly16-1. (**C**) Schematic overview of the action model of Aly16-1. X, M, or G; n, n ≥ 1 and natural number.

## Data Availability

The original contributions presented in the study are included in the article/Appendix A, further inquiries can be directed to the corresponding author.

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
