# Peer review of "Alginate-Degrading Modes, Oligosaccharide-Yielding Properties, and Potential Applications of a Novel Bacterial Multifunctional Enzyme, Aly16-1"

_microorganisms, 2024, doi:10.3390/microorganisms12112374_

Round 1
Reviewer 1 Report
Comments and Suggestions for Authors
The authors of the microorganisms-3261076 manuscript investigate the multifunctional enzymes Aly16-1 of Streptomyces sp. strain CB16. The manuscript needs improvements before being considered for publication.
The main manuscript's shortcoming is related to the lack of substantiation in the Results and Discussion section of one of the claimed objectives, "enzyme usage potentials." More info regarding potential applications (e.g., of oligosaccharides resulting from enzyme action on different acidic polysaccharides) must be added in the Results and Discussion section.
Another issue that authors must fix is related to the information supplied in the caption of the figures. Figures must be standalone from the main text. Therefore, the acronym used in the figures must be included in the figures caption.
The figures are not numbered in the correct order. Figures S1 and S1 (from the supplementary material?) are included in the manuscript body.
The details of the size exclusion HPLC method used for oligo-saccharides separation must be presented.
More info regarding the China patent No. ZL 2021 1 0379188.5, i.e., the main claim, will be helpful for the readers to substantiate potential applications.
The template of the Microorganism journal must be followed – Introduction, Material and Methods, Results, and Discussions.
Author Response
The authors of the microorganisms-3261076 manuscript investigate the multifunctional enzymes Aly16-1 of Streptomyces sp. strain CB16. The manuscript needs improvements before being considered for publication.
Comments 1: The main manuscript's shortcoming is related to the lack of substantiation in the Results and Discussion section of one of the claimed objectives, "enzyme usage potentials." More info regarding potential applications (e.g., of oligosaccharides resulting from enzyme action on different acidic polysaccharides) must be added in the Results and Discussion section.
Answers: Thank you very much!
- In the revised version, we have added the Discussion section and marked in red.
- We have indicated two application ways of the enzyme: A. the direct oligosaccharide preparation of ΔM; B. oligosaccharide sequencing.
Comments 2: Another issue that authors must fix is related to the information supplied in the caption of the figures. Figures must be standalone from the main text. Therefore, the acronym used in the figures must be included in the figures caption. The figures are not numbered in the correct order. Figures S1 and S1 (from the supplementary material?) are included in the manuscript body.
Answers: Thank you very much! We have carefully checked throughout the manuscript and corrected errors, separated them on in-dependent images, marketing in red.
Comments 3: The details of the size exclusion HPLC method used for oligo-saccharides separation must be presented.
Answer: Thanks for the suggestion! We have changed a sentence in the Material @ Method section (original Line 368), into “The HPLC chromatography by a size exclusion column SuperdexTM 30 Increase 10/300 GL (GE) were performed using 0.20 mol/l NH4HCO3 at the speed of 0.40 ml/min.”
Comments 4: More info regarding the China patent No. ZL 2021 1 0379188.5, i.e., the main claim, will be helpful for the readers to substantiate potential applications.
Answer: Thank you very much! In the original line 430, to express clearer, we have corrected as “The multifunctional polysaccharide-depolymerase Aly16-1, with the M-preference against alginate, corresponding oligosaccharide-yielding properties, and potential oligosaccharide preparing and sequencing applications were authorized within the China Invention patent, under the No. ZL 2021 1 0379188.5.”
Comments 5: The template of the Microorganism journal must be followed – Introduction, Material and Methods, Results, and Discussions.
Answer: Thanks for the suggestion. We have applied the template submitted a new version, and we have corrected the encoding numbers of paragraphs (marked in red), figures (in red), and references (in blue) accordingly.
Additionally, two paragraphs were deleted and moved into the discussion section.
- Usually, reported lyases containing a GAG module could act on various polysaccharide substrates, e.g., HA and CS-A, CS-C, CS-D and CS-E types, while only a few was active against other polysaccharide types besides the GAG types (46-49). In this study, although sharing sequence identities < 30% with characterized enzymes of the PL8 family, e.g., GAGs lyases or alginate lyases, the enzyme rAly16-1 showed degradation activities against various polysaccharides in initial DNS-reducing sugar tests (data not shown), and the β-elimination digestions were each further confirmed by 1H-NMR tests (Figure S1). Therefore, the protein Aly16-1 is defined as a novel PL8 member, mainly due to be multifunctional toward to various acidic polysaccharides, particularly against alginate
- “Therefore, the enzyme of Aly16-1 is useful in the direct preparation of the disaccharide ∆M. Furthermore, the final main oligosaccharide products of alginate by the enzyme Aly16-1 have a succession law, originating from large chains with ∆G terminus, till turning into short chains with ∆M terminus, which was similar to Pae-rAlgL and Avi-rAlgL, two M-specific and endolytic alginate lyases (28, 46), whereas novel for its ∆M-yielding.”
Reviewer 2 Report
Comments and Suggestions for Authors
17 October 2024
Manuscript ID: microorganisms-3261076
Title: Alginate-Degrading Modes, Oligosaccharide-Yielding Properties, and
Potential Applications of a Novel Bacterial Multifunctional Enzyme Aly16-1
Authors: Lianghuan Zeng, Junge Li, Jingyan Gu, Wei Hu, Wenjun Han, Yue-zhong Li
The author elucidated Aly16-1 of Streptomyces sp. strain CB16 as a novel multifunction member of the 8th polysaccharide lyase (PL8) family. Experimental results determined that the recombinant enzyme, rAly16-1, is a multifunctional PL active against various acidic polysaccharides, different types of glycosaminoglycan, xanthan, and alginate.
This work offers valuable insights into the catalytic properties and applications of PL8 family enzymes.
There is a gap in the literature concerning the research presented in the manuscript. The methodology is well-defined, and the conclusions align with the evidence and arguments presented.
Additional comments regarding the tables, figures, and the quality of the data and text in the manuscript are presented below:
1. The abstract is unclear due to by excessive use of abbreviations and symbols. Please revise and correct them.
2. Line 18 – Correct the notification of Celsius degree.
3. Line 20 - Check and correct the use of “∆” and “ ≥” symbols by replacing them with words
4. Line 92- Use the Microsoft Word template or LaTeX template to prepare your manuscript. The first section shoul be “Materials and Methods” followed by “Results” section.
5. Ensure the manuscript follows the Microsoft Word template for formatting (Table, Figure, subsection, etc.).
6. Line 93 - Figures should not appear immediately after subsection title (same issue in line 159).
7. Figure S1 and lines 133-138 – Provide full description of the abbreviations: HA, CSA, CSC, CSE, CSD. Are these same as CS-A, CS-C, CS-E, CS-D? Ensure consistency and correct usage.
8. Lines 149-154 - Correct the Celsius degree notation and the way ions are indicated.
9. Figure S2 – Enlarge the figure as the chemical ingredients descriptions in figure D are not visible.
10. Line 177 - Change “20.6’” to “20.6 min”.
11. Review the entire text and replace abbreviations and symbols with their full names where appropriate (ΔM, M3, UM2, M4, M5, M6, 2AB, H-4ΔM, etc.). The excessive use of abbreviations and symbols diminishes the quality of the text
12. Consider adding the following reference: Li, Q., Zheng, L., Guo, Z., Tang, T., & Zhu, B. (2021). Alginate degrading enzymes: an updated comprehensive review of the structure, catalytic mechanism, modification method and applications of alginate lyases. Critical Reviews in Biotechnology, 41(6), 953–968. https://doi.org/10.1080/07388551.2021.1898330
13. Lines 455 – 560 - The references should be formatted according to the journal’s quidelines.
Conclusion: I recommend this manuscript be published in Microorganism after major revision.
Author Response
The author elucidated Aly16-1 of Streptomyces sp. strain CB16 as a novel multifunction member of the 8th polysaccharide lyase (PL8) family. Experimental results determined that the recombinant enzyme, rAly16-1, is a multifunctional PL active against various acidic polysaccharides, different types of glycosaminoglycan, xanthan, and alginate.
This work offers valuable insights into the catalytic properties and applications of PL8 family enzymes.
There is a gap in the literature concerning the research presented in the manuscript. The methodology is well-defined, and the conclusions align with the evidence and arguments presented.
Additional comments regarding the tables, figures, and the quality of the data and text in the manuscript are presented below:
Comments 1:The abstract is unclear due to by excessive use of abbreviations and symbols. Please revise and correct them.
Answer: Thanks for your suggestions. We have made corrections through the abstract part and the whole manuscript, corrected in the revised version.
Comments 2:Line 18 – Correct the notification of Celsius degree.
Answer: Thank you for the suggestion, and each “℃” has been changed into the Palatino Linotype with the size in accordance with the rule, throughout the manuscript and marked in red.
Comments 3:Line 20 - Check and correct the use of “∆” and “ ≥” symbols by replacing them with words
Answer: Thank you for the suggestion. We have corrected as following:
- Line 19, “(∆: unsaturated monosaccharide)” was inserted before “trisaccharide”; and the following sentence was corrected as ”and the minimal product is ∆”.
- Line 20, “fractions ≥ trisaccharide” was changed into “… fractions larger than disaccharides”
- Line 21, “products” was deleted, and “an” was inserted after “being”.
- Line 22, “variable” was corrected as “various”, and “mainly” was deleted.
- Line 23, The words “various” and “which was” have been deleted, and “nr” has been changed into “non-reducing”..
- Line 24, “Therefore” was changed into “Thus”.
- Moreover, please revise the “clean” version.
Comments 4: Line 92- Use the Microsoft Word template or LaTeX template to prepare your manuscript. The first section should be “Materials and Methods” followed by “Results” section. Ensure the manuscript follows the Microsoft Word template for formatting (Table, Figure, subsection, etc.).
Answer: Thanks for your suggestions. We have then used the WORD template and made corrections of each Fig. number (in red), paragraph (in red), and reference indication (in blue) accordingly.
Comments 5: Line 93 - Figures should not appear immediately after subsection title (same issue in line 159).
Answer: Thank you for the suggestion.
- We have moved the position of lines 103-121 from “after Fig. 1” into “in front of Fig. 1” and indicated them in red.
- We have moved the position of lines 165-188from “after Fig. S2” into “in front of Fig. S2” and indicated them in red.
Comments 6: CSE, CSD. Are these same as CS-A, CS-C, CS-E, CS-D? Ensure consistency and correct usage.
Answer: Thank you! We have corrected by using WORD template, and moreover presented each full name in the Materials & Methods section. And it can be seen in the “clean “version, for instance,
1) “CS-A, C, D and E types” in the line 100, meaning polysaccharide substates that are comprised of certain sugar elements instead of enzymes.
2) in the lines 249-250, corrected as “obviously acting on the HA, CS-A, CS-C, and CS-E polysaccharides”.
Comments 7: Lines 149-154 - Correct the Celsius degree notation and the way ions are indicated.
Answer: Thank you! We have checked throughout the manuscript and made corrections of “°C” accordingly. Moreover, we marked them in red.
Comments 8: Figure S2 – Enlarge the figure as the chemical ingredients descriptions in figure D are not visible.
Answer: Thank you for the suggestion! We have changed the width of Fig S2 from 12 cm to 16 cm accordingly, and more separate the FigS2 into an independent in-mage.
Comments 9: Line 177 - Change “20.6’” to “20.6 min”.
Answer: Thank you! Sorry for our careless, and we have made the correction and marked in red.
Comments 10: Review the entire text and replace abbreviations and symbols with their full names where appropriate (ΔM, M3, UM2, M4, M5, M6, 2AB, H-4ΔM, etc.). The excessive use of abbreviations and symbols diminishes the quality of the text
Answer: Thanks for your suggestions! We have checked throughout the manuscript and corrected for instance:
- Line 303, M-enriched disaccharide M2;
- Line 308, “a M-enriched saturated tetrasaccharide”;
- Line 310, “UM3 (an M-enriched and unsaturated trisaccharide)”;
- Line 311, “M5 (an M-enriched and saturated polar pentasaccharide)”;
- Line 350, “unsaturated disaccharide (UDP2) to unsaturated octasaccharide (UDP8)”;
- Line 356, “larger than a UDP2-size”.
Comments 11: Consider adding the following reference: Li, Q., Zheng, L., Guo, Z., Tang, T., & Zhu, B. (2021). Alginate degrading enzymes: an updated comprehensive review of the structure, catalytic mechanism, modification method and applications of alginate lyases. Critical Reviews in Biotechnology, 41(6), 953–968. https://doi.org/10.1080/07388551.2021.1898330
Answer: Thanks for your suggestions! We have updated it as the Reference 24 and marked it in red. Moreover. We have newly formatted the references.
Comments 12: Lines 455 – 560 - The references should be formatted according to the journal’s quidelines.
Answer: Thanks for your suggestions! We have updated References with the emphasis on the rule, and marked in red.
Conclusion: I recommend this manuscript be published in Microorganism after major revision.
Answer: Thank you very much!
Reviewer 3 Report
Comments and Suggestions for Authors
This manuscript studies and partially characterizes a polysaccharide lyase (depolymerase) termed Aly16-1 from Streptomyces sp. strain CB16. The recombinant enzyme is expressed and purified. This is novel multifunctional enzyme of the PL8 family. About substrate specificity, according to me understanding, it is demonstrated that Aly16-1 is an M-preferred enzyme, and the final main unsaturated oligosaccharide product is a disaccharide ΔM from the r end, even when M-enriched saturated substrate chains enlarged their sizes.
The manuscript is interesting, methods are right and well described, and the references are appropriate (but see below). Authors display a high expertise in this field. However, before definitive acceptance, some points should be addressed. They are related to the improvement of the results description to facilitate comprehension. In particular, authors should consider the following points:
Abstract should be comprehensible by itself: M and G are rightly defined as mannuronate (M) and guluronate (G), but other abbreviations are difficult to understand for standard readers. In particular, the monosaccharide Δ should be well defined. Reduced and non-reduced ends (n, nr) would be also defined. An abbreviation list would also be informative to follow the work. This piece of work is written for researchers with expertise in polysaccharide depolymerases (PDs), but it is difficult to follow by regular biotechnologists and biologists. By the way, this study is very related with the recent paper doi: 10.1007/s00216-024-05299-5 entitled Analysis of unsaturated alginate oligosaccharides using high-performance anion exchange chromatography coupled with mass spectrometry, which is not cited.
The potential usage and possible applications of the enzyme in comparison to similar enzymes would be briefly discussed. It is stated just that the exact function of this enzyme Aly16-1 must be discovered (line 116), but some brief speculation or suggestions could be written at the discussion.
A graphical scheme showing the structure of the unsaturated oligosaccharide would be informative. Figure 6 contains some structure, but in my opinion the term “unsaturated” and the symbol Δ is not really well defined to facilitate the correct comprehension of the manuscript.
Letter size at the conclusion should be repaired to be uniform.
Supp. Figures should be taken out from the manuscript and presented as supplementary material. Chemicals at the Figure 2S D are difficult to read. Therefore, the most potent inhibitors (it seems that some cupric or ferrous metal salts) would be emphasized somewhere.
Author Response
This manuscript studies and partially characterizes a polysaccharide lyase (depolymerase) termed Aly16-1 from Streptomyces sp. strain CB16. The recombinant enzyme is expressed and purified. This is novel multifunctional enzyme of the PL8 family. About substrate specificity, according to me understanding, it is demonstrated that Aly16-1 is an M-preferred enzyme, and the final main unsaturated oligosaccharide product is a disaccharide ΔM from the r end, even when M-enriched saturated substrate chains enlarged their sizes.
The manuscript is interesting, methods are right and well described, and the references are appropriate (but see below). Authors display a high expertise in this field. However, before definitive acceptance, some points should be addressed. They are related to the improvement of the results description to facilitate comprehension. In particular, authors should consider the following points:
Comments 1: Abstract should be comprehensible by itself: M and G are rightly defined as mannuronate (M) and guluronate (G), but other abbreviations are difficult to understand for standard readers. In particular, the monosaccharide Δ should be well defined. Reduced and non-reduced ends (n, nr) would be also defined. An abbreviation list would also be informative to follow the work.
Answer: Thank you! In the line 19, “Δ” was newly defined as “(∆: unsaturated monosaccharide)” in the abstract section and was marked in red.
Comments 2: This piece of work is written for researchers with expertise in polysaccharide depolymerases (PDs), but it is difficult to follow by regular biotechnologists and biologists.
Answer: Thanks for your suggestion. To ensure the success of repeat, we declare that all the authors are glad to share all the oligosaccharide samples or enzyme tools with public readers or potential research partners.
Comments 3: By the way, this study is very related with the recent paper doi: 10.1007/s00216-024-05299-5 entitled Analysis of unsaturated alginate oligosaccharides using high-performance anion exchange chromatography coupled with mass spectrometry, which is not cited.
Answer: Thanks for your suggestions! We have updated it as the Reference 42 and marked it in red. And the original efference numbers were updated accordingly
Comments 4: The potential usage and possible applications of the enzyme in comparison to similar enzymes would be briefly discussed.
Answer: Thank you! In order to discuss it further and briefly, we have added a discussion part at the end of the main text body.
Comments 5: It is stated just that the exact function of this enzyme Aly16-1 must be discovered (line 116), but some brief speculation or suggestions could be written at the discussion.
Answer: Thanks! We have used the template and provided a Discussion section in the revised version, in red.
Comments 6: A graphical scheme showing the structure of the unsaturated oligosaccharide would be informative. Figure 6 contains some structure, but in my opinion the term “unsaturated” and the symbol Δ is not really well defined to facilitate the correct comprehension of the manuscript.
Answer: Thank you for the suggestion! Within a natural alginate polysaccharide chain, the unsaturated monosaccharide Δ does not exist, while it was produced in a lyase-reacted system. In order to exhibit the oligosaccharide products, particular the M-preference and the possible cleaving motif inside, we had to use the letter “Δ” together with M and G only as characters within the in-mage.
Comments 7: Letter size at the conclusion should be repaired to be uniform.
Answer: Thank you very much! We have checked and corrected the letter size according to the template.
Comments 8: Supp. Figures should be taken out from the manuscript and presented as supplementary material. Chemicals at the Figure 2S D are difficult to read. Therefore, the most potent inhibitors (it seems that some cupric or ferrous metal salts) would be emphasized somewhere.
Answer: Thank you for the suggestion! The same question and answer to Q9 of Reviewer 2 and:
- Fig. S1 has been provided as an independent in-mage, adding the word “1H-absorbances among products at 5.6 ~ 5.7 ppm suggested the similar degradation via the β-elimination mechanism as a lyase.” in red.
- To enhance the figure quantity, we have changed the width of Fig. S2 from 12 cm to 16 cm accordingly. And, Fig. S2 has been provided as an independent in-mage
Round 2
Reviewer 1 Report
Comments and Suggestions for Authors
The authors made the requested improvements. The manuscript is suitable for publication.
Author Response
Thank you very much!
Reviewer 2 Report
Comments and Suggestions for Authors
5 November 2024
Manuscript ID: microorganisms-3261076 v.2.
Title: Alginate-Degrading Modes, Oligosaccharide-Yielding Properties, and
Potential Applications of a Novel Bacterial Multifunctional Enzyme Aly16-1
Authors: Lianghuan Zeng, Junge Li, Jingyan Gu, Wei Hu, Wenjun Han, Yue-zhong Li
I have reviewed it in detail and have provided the following comments and suggestions to help improve its clarity and readability. Overall, I find the manuscript well-conducted and recommend it for publication in Microorganism after minor revisions. Please consider the following points:
1. Line 182, These lines require checking and any necessary corrections.
2. Lines 253-261 These lines require checking and any necessary corrections.
3. Lines 358-362 These lines require checking and any necessary corrections.
4. Consider adding a list of abbreviations to the manuscript. This will help readers navigate the text more easily.
Review the entire text and replace abbreviations and symbols with their full names where appropriate (ΔM, M3, UM2, M4, M5, M6, 2AB, H-4ΔM, etc.). The excessive use of abbreviations and symbols diminishes the quality of the text
5. References (Lines 484 – 589) - Ensure all references follow the journal's specific formatting guidelines. This includes consistency in style, order, and punctuation.
6. Line 553, These lines require checking and any necessary corrections.
7. Figure S2 – To improve readability, consider presenting the figure in separate parts.
Conclusion: I recommend this manuscript be published in Microorganism after minor revision.
Author Response
Manuscript ID: microorganisms-3261076 v.2.
Title: Alginate-Degrading Modes, Oligosaccharide-Yielding Properties, and
Potential Applications of a Novel Bacterial Multifunctional Enzyme Aly16-1
Authors: Lianghuan Zeng, Junge Li, Jingyan Gu, Wei Hu, Wenjun Han, Yue-zhong Li
I have reviewed it in detail and have provided the following comments and suggestions to help improve its clarity and readability. Overall, I find the manuscript well-conducted and recommend it for publication in Microorganism after minor revisions.
Answer: Thanks a lot!
Please consider the following points:
- Line 182, These lines require checking and any necessary corrections.
Answer: Thank you! We have corrected as “NaH2PO4-Na2HPO4 buffer” and marked it in red.
- Lines 253-261 These lines require checking and any necessary corrections.
Answer: Thanks! We have carefully read the part by correcting into “Ag+, Cu2+, Hg2+, Zn2+, Fe3+” and marked them in red.
- Lines 358-362 These lines require checking and any necessary corrections.
Answer: Thank you! We have carefully read the part by adding “~ 1:1 molar ratio” in the line 362 and marked it in red.
- Consider adding a list of abbreviations to the manuscript. This will help readers navigate the text more easily.
Review the entire text and replace abbreviations and symbols with their full names where appropriate (ΔM, M3, UM2, M4, M5, M6, 2AB, H-4ΔM, etc.). The excessive use of abbreviations and symbols diminishes the quality of the text
Answer: Thanks for your suggestions again! We have read the whole manuscript carefully, adding an “5. Abbreviation” part in the lines 443-454, and re-named “6. Patent” accordingly.
- References (Lines 484 – 589) - Ensure all references follow the journal's specific formatting guidelines. This includes consistency in style, order, and punctuation.
Answer: Thank you! We have read the references in the part carefully, correcting as ruled.
- Line 553, These lines require checking and any necessary corrections.
Answer: Thank you! We have read the reference in the part carefully and corrected as “Analytical and Bioanalytical Biochemistry”, marketing in red.
- Figure S2 – To improve readability, consider presenting the figure in separate parts.
Answer: Thanks for your valuable suggestions. We have submitted a separate figure of Fig. S2 again for publishment and to improve readability.
Conclusion: I recommend this manuscript be published in Microorganism after minor revision.
Answer: Thank you very much!
More:
We have checked through the manuscript in the previous WORD (marked up) version and corrected them as following,
- Line 155, 174, 176, 192, 256, 257, 258 and 191, each letter or word in red has been changed into black.
- Line 427, “proved“ into ”provided“.
- Line 438, “novel sights” into “novel insights”.
- Line 440, “improvement” into “the improvement”.